# Newborn Screening for Acid Sphingomyelinase Deficiency: Prevalence and Genotypic Findings in Italy

**DOI:** 10.3390/ijns10040079

**Published:** 2024-12-04

**Authors:** Vincenza Gragnaniello, Chiara Cazzorla, Daniela Gueraldi, Christian Loro, Elena Porcù, Leonardo Salviati, Alessandro P. Burlina, Alberto B. Burlina

**Affiliations:** 1Division of Inherited Metabolic Diseases, Department of Women’s and Children’s Health, University Hospital of Padua, 35128 Padua, Italy; vincenza.gragnaniello@aopd.veneto.it (V.G.); chiara.cazzorla@aopd.veneto.it (C.C.); daniela.gueraldi@aopd.veneto.it (D.G.); christian.loro@aopd.veneto.it (C.L.);; 2Division of Inherited Metabolic Diseases, Department of Women’s and Children’s Health, University of Padua, 35128 Padua, Italy; 3Clinical Genetics Unit, Department of Women’s and Children’s Health, University of Padua, 35128 Padua, Italy; leonardo.salviati@unipd.it; 4Neurology Unit, St Bassiano Hospital, 36061 Bassano del Grappa, Italy; alessandro.burlina@aulss7.veneto.it

**Keywords:** acid sphingomyelinase deficiency, newborn screening, LysoSM

## Abstract

Acid sphingomyelinase deficiency (ASMD) is a rare lysosomal storage disorder with a broad clinical spectrum. Early diagnosis and initiation of treatment are crucial for improving outcomes, yet the disease often goes undiagnosed due to its rarity and phenotypic heterogeneity. This study aims to evaluate the feasibility and disease incidence of newborn screening (NBS) for ASMD in Italy. Dried blood spot samples from 275,011 newborns were collected between 2015 and 2024 at the Regional Center for Expanded NBS in Padua. Acid sphingomyelinase activity was assayed using tandem mass spectrometry. Deidentified samples with reduced enzyme activity underwent second-tier testing with LysoSM quantification and *SMPD1* gene analysis. Two samples were identified with reduced sphingomyelinase activity and elevated LysoSM levels. Both carried two *SMPD1* variants, suggesting a diagnosis of ASMD. Molecular findings included novel and previously reported variants, some of uncertain significance. The overall incidence was 1 in 137,506 newborns and the PPV was 100%. This study demonstrates the feasibility of NBS for ASMD in Italy and provides evidence of a higher disease incidence than clinically reported, suggesting ASMD is an underdiagnosed condition. Optimized screening algorithms and second-tier biomarker testing can enhance the accuracy of NBS for ASMD. The long-term follow-up of identified cases is necessary for genotype–phenotype correlation and improving patient management.

## 1. Introduction

Acid sphingomyelinase deficiency (ASMD), formerly known as Niemann–Pick disease types A and B, is an autosomal recessive disorder caused by pathogenic variants in the sphingomyelin phosphodiesterase 1 (*SMPD1*) gene. The encoded acid sphingomyelinase enzyme, a lysosomal hydrolase, is crucial for degrading sphingomyelin into ceramide and phosphocholine. A deficiency in this enzyme leads to the progressive accumulation of sphingomyelin (SM) in organs and tissues, prominently affecting the monocyte–macrophage system (spleen, liver, lung, bone marrow) and central nervous system (CNS) neurons [1].

ASMD is a pan-ethnic disorder with an estimated prevalence of 1:250,000 live births and a variable incidence across different ethnicities, reaching up to 1:40,000 live births in the Ashkenazi Jewish population [2,3,4].

The disease presents a wide, continuous clinical spectrum, classically categorized into three forms. The infantile neurovisceral form (ASMD type A, OMIM #257200) manifests during the first months of life with a severe neurodegenerative phenotype and early mortality [5]. The chronic visceral form (ASMD type B, OMIM #607616) presents at variable ages as slowly progressive visceral disease without neurological involvement [6,7]. The chronic neurovisceral ASMD (ASMD type A/B) is an intermediate phenotype occurring during childhood, characterized by progressive somatic and neurological symptoms [8,9].

The diagnosis is confirmed by measuring enzyme activity in dried blood spots (DBS), leukocytes, or fibroblasts, assessing biomarkers (LysoSM and LysoSM509), and conducting genetic analysis [10,11].

Recently, an enzyme replacement therapy (ERT), olipudase alfa, has received regulatory approval in many countries for treating the visceral manifestations of ASMD. It appears to be effective in minimizing disease manifestations and improving outcomes [12,13]. However, diagnostic delays are common due to the heterogeneity of disease presentation, the rarity of the disorder, and a subsequent lack of awareness [14,15].

In a recent Italian study, experts unanimously agreed that patients with ASMD types B and A/B experience delays in diagnosis or misdiagnosis of other conditions, such as metabolic disorders with hepatosplenomegaly, infectious diseases, or malignancies, leading to the risk of inappropriate and potentially harmful treatments. For both types, experts identified diagnostic delay as a major unmet need [16].

These findings are corroborated by international studies. Recently, Doerr et al. analyzed approximately 200 charts of ASMD types B and A/B patients and reported diagnostic delays ranging from 0 to 10 years. On average, patients experience five symptoms and consult three physicians during the diagnostic process. They undergo numerous tests, causing worry, frustration, and weariness. Moreover, only 15% of patients receive an initial ASMD diagnosis, while the remaining 85% receive an incorrect initial diagnosis [17].

Confirming that the incidence of ASMD is underestimated, a recent selective screening study in a high-risk population with clinical suspicion of Gaucher disease demonstrated an incidence of one case of ASMD for every four cases of Gaucher disease [18].

These reasons, along with the availability of new technologies, have led to the consideration of ASMD as a candidate for newborn screening (NBS).

However, to date, the use of NBS for ASMD is scarce due to the only recent availability of an effective therapy and the estimated low incidence of the disease (Table 1).

In the USA, only 4% of states screen for ASMD, compared to 64% for Pompe disease and 60% for MPSI [24]. The largest study was conducted in Illinois, where 1,230,900 newborns were screened between 2014 and 2023. Ten showed low ASMD activity, and all were confirmed by molecular analysis with ASMD (eight ASMD type B, two undetermined phenotypes), with an incidence of 1 in 126,345 [28]. In New York, NBS for ASMD started in 2013, and 65,605 newborns were screened [29]. Two infants were homozygous for different but previously undescribed variants of uncertain significance (VUSs). Finally, in Washington State, a pilot project screened about 43,000 deidentified newborn samples. Five samples showed low ASM activity, but only one carried two pathogenic variants of the *SMPD1* gene [26].

Outside the USA, only small-scale NBS experiences have been reported in South America (Mexico, 20,018 newborns [25]; Brazil, 20,066 neonates [24]) and China (Shandong province, 38,945 neonates [22]), but no newborn was found to be affected by ASMD.

In Europe, two deidentified studies were reported. In Austria, blood spot samples from 34,736 newborns were collected in 2010. One positive sample was found with low enzyme activity for ASM, but this was not confirmed by mutation analyses [19]. In 2012, Wittmann et al. reported the application of the MS/MS method to screen for lysosomal storage disorders (LSDs) in 40,024 deidentified newborn samples from the Hungarian NBS program in Szeged. Abnormally low activity for ASM was found in five samples, which were submitted for molecular diagnosis. In two of them, pathogenic variants predictive of ASMD A/B were found [20]. To our knowledge, there are no previous Italian experiences.

In 2015, the regional health government of north-east Italy included four lysosomal storage diseases (Gaucher disease, Pompe disease, Fabry disease, and Mucopolysaccharidosis I) in the expanded NBS panel. Enzyme activities were assayed in DBS by multiplex tandem mass spectrometry (MS/MS) [30]. ASMD was not included in the official screening program, but data were deidentified and collected.

Here, we report the results of our study to evaluate the feasibility and prevalence of ASMD in Italy and discuss the advantages and disadvantages of this screening approach to guide future policy decisions.

## 2. Materials and Methods

### 2.1. Study Population

Dried blood spot (DBS) samples were consecutively collected from 275,011 newborns between September 2015 and September 2024 at the Regional Center for Expanded Newborn Screening, Padua University Hospital. According to the NBS protocol, samples were obtained between 36 and 48 h after birth, on the same card used for other newborn screening tests. A second sample was required for premature infants (<34 gestational weeks and/or weight < 2000 g) and for sick newborns (those receiving transfusions or parenteral nutrition).

### 2.2. Screening Assay

Enzyme activity was assayed on DBS samples using the NeoLSD^®^ kit from PerkinElmer (Turku, Finland) and liquid chromatography–tandem mass spectrometry (LC–MS/MS). This kit enables the simultaneous determination of six lysosomal enzyme activities: acid β-glucocerebrosidase (Gaucher disease), acid α-glucosidase (Pompe disease), acid α-galactosidase (Fabry disease), acid α-L-iduronidase (Mucopolysaccharidosis I), acid sphingomyelinase (ASMD), and β-galactosidase (Krabbe disease). Data on acid sphingomyelinase (ASM) activity were collected in a deidentified manner.

The cutoff value (0.2 multiple of the median, MoM) was recalculated monthly. Samples with questionable sample integrity (low activities for two or more enzymes) were excluded from further workup.

If the DBS sample showed low ASM activity, the deidentified sample was subjected to a second-tier test (IITT) on the same DBS. This involved quantification of LysoSM using a multiplex assay by LC-MS/MS, as previously described [31]. LysoSM levels above 51.68 nmol/L were considered abnormal.

For these abnormal samples, genotyping was performed to identify potential pathogenic variants in the *SMPD1* gene. Genomic DNA was extracted from the DBS, and next-generation sequencing (NGS) was carried out using the Illumina MiSeq Sequencing System to determine specific exonic regions, as well as exon-intron boundaries. Identified variants were annotated using the human *SMPD1* (NM_000543.5) sequence as a reference. Variants were evaluated for potential pathogenicity using the ClinVar database, Polyphen-2, and SIFT algorithms. The ExAC browser (Beta) (Exome Aggregation Consortium) https://gnomad.broadinstitute.org, accessed on 30 September 2024) was used to compare previously reported variant allele frequencies in the general population. Rare DNA variants were also searched for in publications containing disease-associated mutations. Novel variants were classified according to the ACMG standards and guidelines.

## 3. Results

Between 2015 and 2024, 275,011 newborns were screened, and two DBS samples were found to have reduced ASM activity. Both also showed increased LysoSM levels. These samples were considered highly suspicious for ASMD and underwent molecular analysis (Table 2).

Sample 1 had an ASM activity of 0.53 μmol/L and a LysoSM level of 62.13 nmol/L (normal value <51.68 nmol/L). It carried the p.Tyr369Cys and p.Arg591Cys variants. The p.Tyr369Cys variant is likely pathogenic (ACMG class 4) and has been reported in homozygous state in ASMD type A patients [29]. The p.Arg591Cys variant is a VUS (ACMG class 3) that has been associated with an increased risk of Parkinson’s disease [32].

Sample 2 had an ASM activity of 0.52 μmol/L and a LysoSM level of 63.68 nmol/L (normal value <51.68 nmol/L). It carried the p.Glu411Serfs14 and p.Ser510Phe variants. The p.Glu411Serfs14 variant is novel, has a minor allele frequency (MAF) of 0, and is likely pathogenic (ACMG class 4). The p.Ser510Phe variant has been reported as benign [33] or as a reduced activity variant [26]. However, prediction software (SIFT, Polyphen) suggests a damaging effect on the protein. The effect of this variant in trans with a severe variant is unknown.

The overall incidence of ASMD in this study was 1 in 137,506 newborns.

## 4. Discussion

In this paper, we present the results of a study on deidentified samples to collect objective evidence about the feasibility and efficacy of NBS for ASMD and disease incidence, which can then inform decision-making processes about a nationwide NBS program.

Given the recent development of therapies and methodologies, there is growing interest in NBS for lysosomal diseases. Our previous experience from the NBS program in northeastern Italy suggests that screening for LSDs is feasible, effective, and can be extended to the larger Italian newborn population [30,34].

Enzyme assays on DBS are the most common methods in NBS programs. For ASMD, the MS/MS assay is the method of choice, as it uses a close structural analogue of the natural substrate, sphingomyelin [35]. This is particularly important because the incorporation of a fluorophore into the substrate in fluorometric assays can lead to false-negative results in patients carrying the p.Gln292Lys variant (pseudonormal activity) [36].

MS/MS is a highly multiplexable method. More than 20 years ago, Li et al. first described a multiplex MS/MS screening method, using a cassette of substrates and internal standards to directly quantify severe enzyme activities simultaneously, including ASM (Fabry disease, Gaucher disease, Krabbe disease, ASMD, Pompe disease) [37]. Subsequently, in 2014, Gelb et al. developed a 6-plex test for Fabry, Gaucher, MPS I, Krabbe, ASMD and Pompe diseases [26]. The kit, containing a buffer, substrates, and internal standards for multiplex assays, was commercialized by PerkinElmer Corp. (NeoLSD^®^, Shelton, CT, USA) and was actually used in our study and several other NBS programs [22,23,24,26]. This test can be easily integrated into screening laboratories that use tandem mass spectrometry for screening lysosomal diseases or other inborn errors of metabolism.

As a cutoff, it is possible to use a fixed enzyme activity cutoff value based on a pre-pilot population analysis or a multiple or percentage of the median or mean enzyme activity. We chose to use the 0.2 multiple of the median (MoM). Interestingly, we noted that the ASMD enzyme activity cutoff value differed according to the season (Figure 1), being lowest in July and August and highest in December and January. These results agree with our previous experience with other lysosomal enzymes [38] and with the results of Li et al., who demonstrated that enzyme activity appears to be affected by temperature, making it difficult to establish a fixed cutoff value [22]. In our study, the cutoff (0.2 MoM) was recalculated monthly to avoid an increase in false positives in winter and false negatives in summer.

The versatility of MS/MS enables this technique to be applied not only for enzymatic determination but also for biomarker detection.

In our study, we used the LysoSM assay by LC–MS/MS as IITT, achieving a positive predictive value (PPV) of 100%.

Plasma LysoSM has long been used as a specific biomarker for the diagnosis and monitoring of ASMD. Recently, we developed a method for the simultaneous quantification of LysoSM and other lysosphingolipids on DBS, which is already used as IITT for other screened diseases (LysoGb3 for Fabry disease, LysoGb1 for Gaucher disease) [39]. Although the determination of LysoSM in DBS seems to be less discriminative than in plasma [31,40,41], in several studies, LysoSM levels in DBS from ASMD patients were found to be substantially elevated (approximately 5 times) when compared to normal controls, with no overlap [31,40,41,42]. However, data on patients in the neonatal period are lacking. LysoSM as IITT was used in only one other study, in Brazil, on a small population (20,066 newborns), but no ASMD patients were identified [41]. In our study, both samples with reduced ASM activity already had elevated LysoSM levels in the neonatal period. This not only increases the specificity of the screening but also allows for better characterization of VUS. Additionally, measuring LysoSM has advantages over molecular testing in terms of cost-effectiveness, time, and the need for specialized expertise [43]. It can be hypothesized that the use of LysoSM IITT will become increasingly widespread in the coming years. In New York, ScreenPlus is a pilot NBS program that aims to enroll over 100,000 infants over a five-year period. This panel includes 14 disorders, including ASMD, and uses an analyte-based, multi-tiered screening platform to enhance screening accuracy. First-tier screening is enzyme-based using a megaplex LC-MS/MS assay. Infants who have an abnormal screen for ASMD on the first-tier assay have their DBS sent for second-tier (LysoSM) and third-tier testing (*SMPD1*) [44].

The use of LysoSM as a IITT could also assist in phenotype prediction. Indeed, with existing clinical and laboratory tools, distinguishing infantile and chronic ASMD patients in early infancy is challenging. The amount of residual enzymatic activity may overlap across the spectrum of ASMD. Phenotype prediction based on genotype can also present challenges. While the presence of nonsense variants, large deletions, or variants leading to a reading frameshift in both alleles is associated with the severe neurovisceral phenotype, establishing a correlation with a specific clinical phenotype is more difficult for pathogenic missense variants or VUS [1,45]. Phenotype predictions may be unreliable due to VUS and unique compound heterozygous combinations, as reported in our and other NBS studies [20,29]. Conversely, Breylin et al. demonstrated that LysoSM elevations in patients with infantile ASMD are greater than those with chronic ASMD, and among patients with chronic ASMD, a positive relationship was observed between LysoSM levels and clinical severity [46]. However, further data are needed about the use of LysoSM levels in the neonatal period and predicting phenotypes in pre-symptomatic individuals.

Data on ASMD epidemiology, including birth prevalence and the frequency of different ASMD phenotypes, are scarce. Available Italian data on incidence are limited to a national retrospective survey of inborn errors of metabolism conducted between 1985 and 1997, which identified 13 type A and 8 type B ASMD cases out of a total of over 7 million live births [47]. In a recent Italian Delphi consensus, the majority of panelists indicated an estimated number of living patients with ASMD in Italy of between 20 and 40. Approximately 70% of ASMD cases are estimated to be chronic visceral (type B), 25% chronic neurovisceral (type A/B), and 5% infantile neurovisceral (type A), consistent with the mortality distribution of the three ASMD phenotypes. Moreover, the majority of experts agreed that for types A and A/B, the percentage of undiagnosed patients is up to 30%, and for type B ASMD, it is between 40% and 80% [16]. Our experience on a large population shows an incidence of 1 in 137,506, higher than previously clinically reported, confirming that ASMD is an underdiagnosed disease. This incidence is comparable to the largest program reported until now, in Illinois, on more than 1,200,000 screened newborns, which provided evidence of a disease incidence of 1 in 1:126,345 [27]. In other NBS studies, the disease incidence is difficult to evaluate due to the small study population compared to the disease prevalence. Despite being rare, the ASMD incidence is similar to other screened lysosomal storage diseases, such as Mucopolysaccharidosis I.

A limitation of our study is the lack of follow-up data. Because our approach eliminates the connection between the sample and the patient, results from our anonymous NBS study are not reported to families, clinical diagnoses and outcome data are not collected, and there is no way to confirm the PPV of the assay. However, although the patient samples were deidentified, we provide genotype data on screen-positive samples to provide an estimate of the number of affected individuals. Thus, our definition of “true positive” includes all phenotypes, regardless of severity and predicted age of onset, and is based on genotypic and biochemical definitions of disease, rather than a definition based on clinical manifestations. However, this is the most practical way to discuss NBS results for disorders with variable ages of presentation or deidentified studies.

### Advantages and Disadvantages

Based on our study and previous experiences, we discuss the advantages and disadvantages of NBS for ASMD.

At first glance, ASMD is ideally suited for NBS; as its markers are readily detectable on DBS, it presents a pediatric phenotype with significant morbidity or mortality if untreated, and a treatment is approved. Early diagnosis through NBS may eliminate the “diagnostic odyssey” experienced by many patients after the onset of symptoms, allowing timely treatment when clinical manifestations first appear [27]. Apart from the potential clinical benefit for patients, neonatal screening can provide information on reproductive risk for parents and future adults and identify additional at-risk or affected family members [48].

Principal limitations of NBS for ASMD are due to:Variants of uncertain significance: VUS and previously unreported variants can make the prediction of phenotypic severity and age of onset challenging. The uncertainty may provoke anxiety for parents as well as healthcare providers [48]. Biomarkers could be useful in determining the pathogenicity of a VUS in the presymptomatic phase. In our experience, LysoSM values are already elevated at birth.Late-onset forms: Current screening methods are unable to distinguish between early and later-onset phenotypes. This means that individuals at risk for later- or adult-onset ASMD may be diagnosed shortly after birth [48]. The onset of signs or symptoms for the late-onset subtypes is variable, and the initiation of treatment depends on the occurrence of the first signs and symptoms [1], so there is a risk of needless anxiety and unnecessary medical intervention and stigmatization in patients who may remain asymptomatic until adulthood [49].Neurologic form: The severe, rapidly progressive neurodegenerative manifestations typical of ASMD type A are not amenable to current therapies, which are unable to cross the blood–brain barrier, so treatment for ASMD type A is limited to supportive therapy [29,48].

## 5. Conclusions and Future Directions

This study demonstrates the feasibility of NBS for ASMD and provides evidence about the disease incidence on more than 250,000 screened newborns in our country. This appears to be higher than the rate clinically estimated in our country but compatible with other screening studies, confirming that ASMD is an underdiagnosed disease. Our approach demonstrates that optimized seasonal cutoff values combined with a two-tier approach could largely reduce the false positive rate.

A limitation of our study is that the data are collected anonymously and are not reported to the patients and families. In the future, before starting a nationwide newborn screening project, it is necessary to develop follow-up protocols for the evaluation of infants with positive screen tests.

The detection of patients carrying adult-onset variants or VUS should be considered. Long-term follow-up programs of these patients will allow better functional characterization of the VUS, elucidate the role of biomarkers, and improve the correlation between genotype and phenotype, thereby enhancing phenotype prediction and optimizing patient management and treatment.

## Figures and Tables

**Figure 1 IJNS-10-00079-f001:**
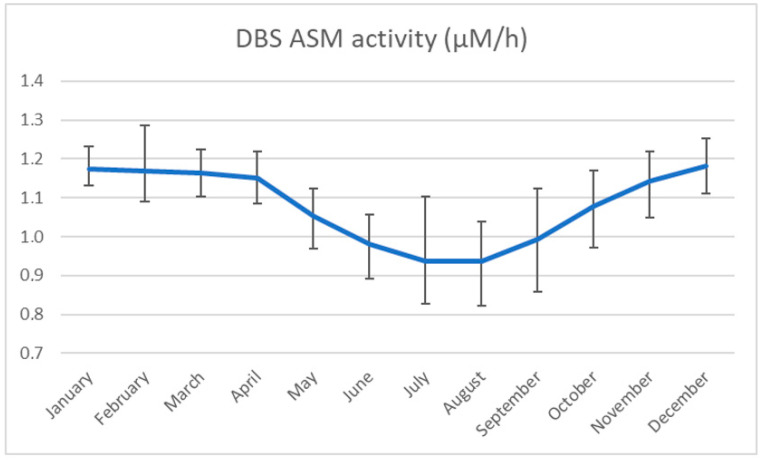
Seasonal variation of ASM activity cutoff in the last 9 years (0.2 MOM, mean and SD).

**Table 1 IJNS-10-00079-t001:** Summary of published NBS programs for ASMD.

Region	Study Period	Numbers of NBS Samples	Method	Cutoff	Second Tier Test	Positive NBS	Confirmed Patients *	Incidence	PPV	Other Screened Diseases
Europe										
Italy (present study)	2015–2023	275,011 deidentified	MS/MS	MOM	DBS LysoSM	2	2	1:137,506	100%	PD, FD, MPSI, GD, KD
Austria [19]	2010	34,736 deidentified	MS/MS	Fixed cutoff	/	1	0	/	/	PD, FD, GD
Hungary [20]	2012	40,024 deidentified	MS/MS	Fixed cutoff	/	5	2	1:20,012	40%	PD, FD, GD
Asia										
Shangai (China) [21]	2021	50,018	MS/MS	MOM	Genetic test	6 (5 genetic test)	5			PD, FD, MPSI, KD, GD
Shandong (China) [22]	2019–2021	38,946	MS/MS	MOM	/	1	0	/	/	PD, FD, MPSI, KD, GD
Fuzhou (China) [23]	2020–2021	3,000	MS/MS	MOM	/	0	0	/	/	PD, FD, MPSI, KD, GD
South America										
Brazil [24]	2021–2023	20,066	MS/MS	%DMA	DBS LysoSM	0	0	/	/	PD, FD, MPSI, KD, GD
Mexico [25]	2012–2016	20,018	MS/MS	Fixed cutoff	/	2	0	/	/	PD, FD, MPSI, GD, KD
USA										
Washington State [26]	2016	43,000 deidentified	MS/MS	%DMA	/	5	1	1:43,000	20%	PD, FD, MPSI, GD, KD
Illinois [27,28]	2014–2023	1,230,900	MS/MS	%DMA	/	10	10	1:126,345	100%	PD, FD, MPSI, GD
New York [29]	2013–2017	65,605	MS/MS	%DMA	/	2	2	1:32,803	100%	PD, FD, MPSI, GD

Abbreviations: PD: Pompe disease, FD: Fabry disease, MPS: mucopolysaccharidosis, KD: Krabbe disease, GD: Gaucher disease, MS/MS: tandem mass spectrometry, DMA: daily mean activity, MOM: multiple of median, PPV: positive predictive value. * Deidentified projects: DNA sequencing if positive.

**Table 2 IJNS-10-00079-t002:** Biochemical and mutational analysis of patients identified by NBS in north-east Italy (2015–2024).

Pt	Yr	Gender	Ethnic Origin	NBS ASM Activity µM/h	DBS lysoSM nMol/L (nv < 51.68)	Gene Variants	Protein Variants	Predicted Phenotype
1	2018	M	Europe	0.53	62.13	c.1106A>G+c.1771C>T	p.Tyr369Cys+p.Arg591Cys	ASMD type B
2	2023	M	Asia	0.52	63.68	c.1231delG+c.1529>T	p.Glu411Serfs*14+p.Ser510Phe	ASMD type B

Pt: patient; Yr: year of birth.

## Data Availability

Data is available on request due to privacy restrictions.

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
