# Peer review of "Newborn Screening for Acid Sphingomyelinase Deficiency: Prevalence and Genotypic Findings in Italy"

_2409-515X, 2024, doi:10.3390/ijns10040079_

Round 1
Reviewer 1 Report
Comments and Suggestions for Authors
This study describe a pilot study for newborn screening for acid sphingomyelinase deficiency in Italy. The authors identified two subjects with biochemical markers of the disease (lower enzymatic values and elevated LysoSM levels) and SMPD1 variants.
This is an important and well written study.
Particularly important is the section on advantages and disadvantages of NBS for this disease.
Specific points:
In the introduction the authors are advised to:
1) Line 42 to 45: The authors should add additional references when describing estimated prevalence of disease.
2) Line 54 to 56: The authors state that “The diagnosis is confirmed by measuring enzyme activity in dried blood spots (DBS), leukocytes, or fibroblasts, assessing substrate accumulation (LysoSM and LysoSM509), and conducting genetic analysis.” This sentence should be re-written as LysoSM and LysoSM509 are not main subtracts of ASM enzyme but results from the partial diacylation of the ASM enzyme subtract sphingomyelin.
3) Line 58 to 60: The authors state: “It should be initiated promptly upon early symptom presentation to prevent irreversible pathological changes, avoid or significantly minimize disease manifestations, and improve outcomes [10,11].” The authors are advised to reformulate this sentence. The available literature, that the authors cite, indicates the positive effects of the therapy in ameliorate disease, however more data is still need to know how early a patient should start therapy and whether the starting age should be the same for patients with different mutations.
Author Response
This study describe a pilot study for newborn screening for acid sphingomyelinase deficiency in Italy. The authors identified two subjects with biochemical markers of the disease (lower enzymatic values and elevated LysoSM levels) and SMPD1 variants.
This is an important and well written study.
Particularly important is the section on advantages and disadvantages of NBS for this disease.
Specific points:
In the introduction the authors are advised to:
1) Line 42 to 45: The authors should add additional references when describing estimated prevalence of disease.
R: We thank the reviewer for the comment. We have added the references 3 (Meikle PJ, Hopwood JJ, Clague AE, Carey WF. Prevalence of lysosomal storage disorders. JAMA. 1999 Jan 20;281(3):249-54. doi: 10.1001/jama.281.3.249) and 4 (Kingma SD, Bodamer OA, Wijburg FA. Epidemiology and diagnosis of lysosomal storage disorders; challenges of screening. Best Pract Res Clin Endocrinol Metab. 2015 Mar;29(2):145-57. doi: 10.1016/j.beem.2014.08.004).
2) Line 54 to 56: The authors state that “The diagnosis is confirmed by measuring enzyme activity in dried blood spots (DBS), leukocytes, or fibroblasts, assessing substrate accumulation (LysoSM and LysoSM509), and conducting genetic analysis.” This sentence should be re-written as LysoSM and LysoSM509 are not main subtracts of ASM enzyme but results from the partial diacylation of the ASM enzyme subtract sphingomyelin.
R: We thank the reviewer for the evaluable comment. We changed “substrate accumulation” with “biomarkers”.
3) Line 58 to 60: The authors state: “It should be initiated promptly upon early symptom presentation to prevent irreversible pathological changes, avoid or significantly minimize disease manifestations, and improve outcomes [10,11].” The authors are advised to reformulate this sentence. The available literature, that the authors cite, indicates the positive effects of the therapy in ameliorate disease, however more data is still need to know how early a patient should start therapy and whether the starting age should be the same for patients with different mutations.
R: We thank the reviewer for the important comment. We have modified the sentence as follow: “It appears to be effective in minimizing disease manifestations, and improving outcomes”.
Reviewer 2 Report
Comments and Suggestions for Authors
This pilot study beautifully demonstrates the feasibility of ASMD NBS on more than 250,000 screened newborns in Italy. The data in this study provided evidence about the ASMD incidence higher than the rate clinically estimated, confirming that ASMD is an underdiagnosed disease. The author described study and methodology clearly, with nice flow and discussion.
I highly recommend this paper for publication with few very minor comments:
1. Author could further refer (in introduction) recent evidence of high ASMD frequency in Gaucher disease high risk population screen. This study reveals that one in four cases suspected for Gaucher disease is diagnosed with ASMD (Mol Genet Metab. 2023 May;139(1):107563).
2. Would be very interesting to know how many samples from 275,011 newborns had low ASM enzyme activity and were selected for second tier Lyso-SPM testing.
3. Would be very interesting to know how many samples from the second tier had elevated Lyso-SPM and were submitted for genetic confirmation. And what was the distribution of Lyso-SPM in this cohort?
Author Response
This pilot study beautifully demonstrates the feasibility of ASMD NBS on more than 250,000 screened newborns in Italy. The data in this study provided evidence about the ASMD incidence higher than the rate clinically estimated, confirming that ASMD is an underdiagnosed disease. The author described study and methodology clearly, with nice flow and discussion.
I highly recommend this paper for publication with few very minor comments:
- Author could further refer (in introduction) recent evidence of high ASMD frequency in Gaucher disease high risk population screen. This study reveals that one in four cases suspected for Gaucher disease is diagnosed with ASMD (Mol Genet Metab. 2023 May;139(1):107563).
R: We thank the reviewer for the important suggestion. We have added the reference in the text (“Confirming that the incidence of ASMD is underestimated, a recent selective screening study in a high-risk population with clinical suspicion of Gaucher disease demonstrated an incidence of 1 case of ASMD for every 4 cases of Gaucher disease”).
- Would be very interesting to know how many samples from 275,011 newborns had low ASM enzyme activity and were selected for second tier Lyso-SPM testing.
- Would be very interesting to know how many samples from the second tier had elevated Lyso-SPM and were submitted for genetic confirmation. And what was the distribution of Lyso-SPM in this cohort?
R: We thank the reviewer for the comment 2 and 3. Two dried blood spots showed low ASM enzyme activity, and both were confirmed with elevated LysoSM levels and two variants in the SMPD1 gene. We have added these data in the text (line 159).